# Fabrication and Evaluation of Polyvinyl Alcohol/Corn Starch/Patchouli Oil Hydrogel Films Loaded with Silver Nanoparticles Biosynthesized in *Pogostemon cablin* Benth Leaves’ Extract

**DOI:** 10.3390/molecules28052020

**Published:** 2023-02-21

**Authors:** Khairan Khairan, Miftahul Hasan, Rinaldi Idroes, Muhammad Diah

**Affiliations:** 1Departement of Pharmacy, Universitas Syiah Kuala, Banda Aceh 23111, Indonesia; 2Departement of Chemistry, Universitas Syiah Kuala, Banda Aceh 23111, Indonesia; 3Herbal Medicine Research Center, Universitas Syiah Kuala, Banda Aceh 23111, Indonesia; 4Ethnoscience Research Center, Universitas Syiah Kuala, Banda Aceh 23111, Indonesia; 5PUI-Nilam Aceh-Atsiri Research Centre, Universitas Syiah Kuala, Banda Aceh 23111, Indonesia; 6Faculty of Medicine, Universitas Syiah Kuala, Banda Aceh 23111, Indonesia; 7Division of Cardiology, Zainoel Abidin Hospital, Banda Aceh 23111, Indonesia

**Keywords:** hydrogel films, silver nanoparticles, *Pogostemon cablin* Benth, *Staphylococcus aureus*, *Staphylococcus epidermis*

## Abstract

Research on the manufacture of hydrogel films from polyvinyl alcohol, corn starch, patchouli oil, and silver nanoparticles, (PVA/CS/PO/AgNPs, respectively) was completed. The silver nanoparticles used in this study resulted from green synthesis using local patchouli plants (*Pogostemon cablin* Benth). Aqueous patchouli leaf extract (APLE) and methanol patchouli leaf extract (MPLE) are used in the synthesis of phytochemicals (green synthesis), which are then blended in the production of PVA/CS/PO/AgNPs hydrogel films, which are then cross linked with glutaraldehyde. The results demonstrated that the hydrogel film was flexible, easy to fold, and free of holes and air bubbles. The presence of hydrogen bonds between the functional groups of PVA, CS, and PO was revealed using FTIR spectroscopy. SEM analysis revealed that the hydrogel film was slightly agglomerated and did not exhibit cracking or pinholes. The analysis of pH, spreadability, gel fraction, and swelling index showed that the resulting PVA/CS/PO/AgNP hydrogel films met expected standards except for the organoleptic properties of the resulting colors, which tended to be slightly darker in color. The formula with silver nanoparticles synthesized in methanolic of patchouli leaf extract (AgMENPs) had the highest thermal stability compared to hydrogel films with silver nanoparticles synthesized in aqueous of patchouli leaf extract (AgAENPs). The hydrogel films can be safely used up to 200 °C. The antibacterial studies revealed that the films inhibited the growth of both *Staphylococcus aureus* and *Staphylococcus epidermis*, as determined by the disc diffusion method, with the best antibacterial activity being against *Staphylococcus aureus*. In conclusion, the hydrogel film F_1_, loaded with silver nanoparticles biosynthesized in aqueous of patchouli leave extract (AgAENPs) and light fraction of patchouli oil (LFoPO) performed the best activity against both *Staphylococcus aureus* and *Staphylococcus epidermis*.

## 1. Introduction

Traditional medicine has been widely used in Indonesia for centuries to maintain health and prevent disease because it has several advantages over synthetic drugs. Many traditional medicines use plant ingredients, and their use and application are based on personal experience [1]. One type of plant that can be used as a traditional medicine for wound treatment, such as cuts and burns, is patchouli (*Pogostemon cablin* (Blanco) Benth), also known as the patchouli plant. The patchouli plant is a plant from the Lamiaceae family. *Pogostemon cablin* (*P. cablin*) is an aromatic plant (an aromatic herb) that generally grows on a commercial scale in India, Malaysia, Philippines, and Indonesia [2]. *P. cablin* is an aromatic shrub with smooth leaves and rectangular stems. The dried leaves of this plant are distilled to produce patchouli oil, which is widely used in various industrial processes. Patchouli alcohol, the main component of patchouli oil, acts as an anti-inflammatory agent and decongestant [3].

Hydrogel is a cross-linked hydrophilic polymer that is highly absorbent with well defined structures. The hydrogel can be used in many applications, especially in the biomedical area. Hydrogel is transparent, supple, adaptable, and non-irritating to the skin. The hydrogel can be used to manage wound healing since it has better compound release (transport media) qualities than other comparable preparations [4]. Hydrogels can be prepared from natural or synthetic polymers or by adding both (natural and synthetic polymers) to produce the best properties. Hydrogel wound dressings have been developed based on polyvinyl alcohol (PVA), chitosan, and starch [5]. Polyvinyl alcohol is a synthetic polymer that is commonly used in hydrogels. Chitosan is a naturally occurring polysaccharide that is biodegradable, biocompatible, non-toxic, and capable of forming films. Starch is a natural polymer with strong cross-linking properties via a hydroxyl groups (OH) [6].

Hydrogels have unique properties such as hydrophilicity, biodegradability, and biocompatibility, which make them ideal for biomedical applications such as drug delivery, tissue engineering, and wound dressing. Because they are non-toxic, biodegradable, antimicrobial (to prevent wound infection), able to absorb wound exudate, and able to regulate environmental humidity, hydrogel films are an ideal material for wound dressings. Hydrogel films composed of PVA and starch (corn starch) have demonstrated good water absorption, film-forming behavior, and mechanical strength [7].

Research in nanoparticles, especially metal nanoparticles, is increasingly attracting attention because of their unique properties, such as catalyst and optical properties, and their potential applications in the health sector [8]. Among the metal nanoparticles that has been synthesized and applied in medicine are silver nanoparticles (AgNPs). Recently a metal nanoparticle synthesis technique has been developed that is simpler, more cost-effective, efficient, and environmentally friendly. This new method uses plant extracts or what is also known as a green synthesis of nanoparticles. The reason plant extracts are used in the green synthesis of silver nanoparticles is because plants contain secondary metabolites such as alkaloids, tannins, steroids, phenolics, saponins, and flavonoids. These metabolites are known to be able to reduce silver ions (Ag^+^) to (Ag^0^) and form silver nanoparticles [9,10]. 

According to Parmar et al., the hybridization of nanoparticles and patchouli oil can aid in forming silver nanoparticles that act as a capping agent of silver nanoparticles. Another advantage of the patchouli oil in the hybrid method in green synthesis of silver nanoparticles is that it is capable of protecting the nanoparticles from degenerative changes such as agglomeration and oxidation. Plant extracts, in this case, patchouli leaves (*P. cablin*) and patchouli oil, can hybridize with metal in the nanoparticle structure to produce synergic effects and increase antibacterial activity. A previous study by researcher showed that patchouli oil concentrations of 5%, 7.5%, and 10% in a spray gel form demonstrated wound-healing effects on male rabbits (*Oryctolagus cuniculus*) [11,12]. Herowati et al., demonstrated that the percentage of wound healing increased when concentrations of patchouli oil were also increased, the higher the concentration of patchouli, the faster the wound healed [12]. Gaikwad also showed that nanogel preparations loaded with silver nanoparticles at a concentration of 0.5 mg/g carbopol can cause superior wound healing effects in mice [13]. Mojally et al. succeeded in making a polyvinyl alcohol/corn starch/castor oil (PVA/CS/CO) hydrogel film loaded with silver nanoparticles biosynthesized using mint leaf extract (*Metha piperita*) [7]. The results of research conducted by Mojally showed that the addition of castor oil and silver nanoparticles in hydrogel film preparations produced better hydrogel film preparations and antibacterial activity against *Staphylococcus aureus* ATCC and *Pseudomonas aeruginosa* ATCC. 

The proposed research aimed to (i) utilize the potential of *Pogostemon cablin* Benth plants in the biosynthesis of AgNP, (ii) develop transparent and biodegradable nanocomposite hydrogel films using PVA/CS/PO hydrogel film loaded with biosynthesized silver nanoparticles, (iii) evaluate the properties of hydrogel films obtained from the hybridisation of PVA/CS/PO/hydrogel film loaded with silver nanoparticles resulting from green synthesis using *Pogostemon cablin* Benth, and (iv) determine the activity of the PVA/CS/PO/loaded with silver nanoparticles against *Staphylococcus aureus* ATCC and *Staphylococcus epidermidis* ATCC.

## 2. Results

### 2.1. Patchouli Leaf Extract

The aqueous patchouli leaf extract (APLE) and methanolic patchouli leaves extract (MPLE) can be seen in Figure 1A below, while the phytochemicals test results of APLE and MPLE patchouli aqueous extracts can be seen in Table 1. 

The phytochemical screening was performed on APLE and MPLE which included alkaloids, flavonoids, tannins, saponins, and steroids. Phytochemical screening aims to determine the secondary metabolites contained in APLE and MPLE. The chemical screening assay revealed that MPLE contained alkaloids, flavonoids, tannins, and terpenoids. Meanwhile, APLE contains alkaloids, tannins, and terpenoids. Table 1 also showed that both extracts (APLE and MPLE) do not contain saponins.

The previous study showed that *P. cablin* extract contains several secondary metabolites, such as monoterpenoids, triterpenoids, sesquiterpenoids, phytosterols, flavonoids, lignin, glycosides, alcohols, and aldehydes [2]. Some researchers reported that the primary and secondary metabolites of *P. cablin* were patchouli alcohol, α-patchoulene, β-patchoulene, αbulnesene, seychellene, norpatchoulenol, pogostone, eugenol, and pogostol [2,14]. Swamy and Sinniah also reported that the alkaloid and flavonoid compounds contained in *P. cablin* played an active role in the pharmacological activities such as antibacterial, antifungal and antiviral. Therefore, *P. cablin* is being explored broadly to obtain main key chemical compounds for discovering new drug compound with great therapeutic potential [2].

#### 2.1.1. FT-IR Analysis of Aqueous Patchouli Leaf Extract (APPLE) and Methanolic Patchouli Leaf Extract (MPLE)

Aqueous patchouli leaves extract (APLE) and methanolic patchouli leaves extract (MPLE) were subjected to FTIR analysis between 400 cm^−1^ and 4000 cm^−1^.

Based on their molecular vibrations, the analysis aims to pinpoint the presence of functional groups in the sample. The analysis results showed that the samples tested contained several functional groups such as C-H, O-H, and C=C. The results showed that APLE had three dominant wavelength points, namely 3400.22 cm^−1^, 2138.15 cm^−1^ and 1640.20 cm^−1^ indicating the presence of functional groups of -O-H, -C-N, and -C=C. After reacting with the silver nitrate (AgNO_3_), a precursor used to synthesize of silver nanoparticle (AgAENPs) showed two dominant wavelength points of 3590.12 cm^−1^ and 1675.11 cm^−1^, indicating the presences of -O-H, and -C=C stretching, respectively (Figure 2A,B and Table 2). Besides that, the spectrum FT-IR of MPLE before reaction exhibited six dominant wavelength points as shown in Table 2. Meanwhile, the spectrum FT-IR of MPLE after reaction in synthesis of silver nanoparticles (AgMENPs) exhibited four dominant wavelength points of 3450.14 cm^−1^; 2200.11 cm^−1^; 1670.21 cm^−1^; and 1150.10 cm^−1^ corresponds to -O-H; -C-N; -C=C; and -C-O stretching respectively (Figure 2A,B and Table 2).

#### 2.1.2. GC-MS Analysis of Aqueous Patchouli Leaves Extract (APLE) and Methanolic Patchouli Leaves Extract (MPLE)

Figure 3 displays the spectrum of GC-MS of aqueous patchouli leaf extract (APLE) and methanolic patchouli leaf extract (MPLE). The results showed that aqueous patchouli leaf extract (APLE) and methanolic patchouli leaf extract (MPLE) contain 71.36% and 30.91% patchouli alcohol, respectively (Figure 3A,B).

These findings indicate that APLE contains higher concentrations of patchouli alcohol than MPLE. Table 3 showed the chemical compositions of APLE and MPLE by analysis of GC-MS. The GC-MS analysis revealed that APLE contained 5 compounds, while MPLE contained 17 compounds. The results also showed that the main compounds contained in APLE were α-guaiene (8.25%), azulene (10.67%), and patchouli alcohol (71.36%). Meanwhile, the main compound contained in MPLE were α-guaiene (20.44%), seychellene (12.66%), α-patchoulene (13.38%), and patchouli alcohol (30.91%).

Khairan et al., reported that the main compound in the extract of *P. cablin* were α-guaiene, α-patchoulene, α-bulnesene, and patchouli alcohol. The GC-MS analysis of patchouli oil extract (an essential oil of *Pogostemon cablin*) contained 21 compounds, 18 of them were sesquiterpenes, and the remaining three of them were sesquiterpenes oxide. Supawan also stated that patchouli alcohol was the main component of the patchouli oil extract, followed by the compound of germacrene A [14,15].

### 2.2. Patchouli Oil Fractionations

The fractionations of the patchouli oil were conducted using a vacuum rotary evaporator (Heidolph, Schwabach, Germany). The fractionation process aims to purify and improve the quality of patchouli oil, so as to increase the concentration of patchouli alcohol. In this study, the fractionations process was performed in two steps, the first was to obtain light fraction of patchouli oil, LFoPO, and the second was to obtain high fraction of patchouli oil, HFoPO (Figure 1B).

First, the fractionation was conducted at an initial temperature of between 60 °C and 65 °C in 2 mlbar pressure. In order to prevent bumping, the temperature was gradually raised every 5–10 min until the temperature reached to 125 °C. At the end of the process the light fraction of the patchouli oil was obtained. Second, to obtain a high fraction of patchouli oil (HFoPO), the fractionation was continued until the temperature reached between 115 °C–160 °C. Table 4 shows the yield percentage of the light fraction (LFoPO) and high fraction (HFoPO) of patchouli oil. Table 4 shows that, the percentage of patchouli alcohol contain in LFoPO and HFoPO were 28.68% and 60.66% respectively (Figure 4). The result also showed that the percentage loss of the oil during the fractionation process of the patchouli oil was 5%. This finding indicates that economically speaking the percentage of oil loss during the fractionation process is not indemnifying.

#### GC-MS LFoPO and HFoPO Analysis

Figure 4 displays the spectrum GC-MS of the fractionated high fraction (HFoPO) and light fraction (LFoPO) patchouli oil. Patchouli alcohol (PA) is the most abundant compound in patchouli oil. Patchouli alcohol is known to have anti-influenza, anti-inflammatory, neuroprotective, anti-depressant properties [15]. Patchouli alcohol is also known to have anticancer properties because it can inhibit the growth of cancer cells through the mechanism of inhibiting HDAC2 enzyme expression, downregulation of c-MYC, and activation of the NF-κB pathway [16].

Astuti et al., stated that the high and light fractions of patchouli oil produced by fractionation had the best dopamine-elevating/antidepressant effects. According to other studies, patchouli alcohol can also accelerate the healing of wounds by reducing inflammation caused by AMPK and TGFb1. A novel approach to treating obesity, insulin resistance, and wound healing may involve patchouli alcohol [17].

### 2.3. Green Synthesis Silver Nanoparticles

Nine milliliters of a 0.1 M AgNO_3_ solution was combined with 1 mL of each APLE and MPLE. After that, the mixture was given two hours to react at room temperature. The color changed from green to dark brown during biosynthesis indicating the formation of silver nanoparticles (AgNPs). Figure 1C,D show the results for silver nanoparticles synthesized in aqueous patchouli leave extract (AgAENPs) and silver nanoparticles synthesized in methanolic patchouli leave extract (AgMENPs) solutions respectively. The results show that green synthesis of silver nanoparticles using methanol extract of *Pogostemon cablin* Benth leaves (AgMENPs) produced a clearer suspension of silver nanoparticles compared to the nanoparticle suspension resulting from green synthesis using aqueous extract of *Pogostemon cablin* Benth leaves (AgAENPs).

#### 2.3.1. UV-Vis Spectroscopy Analysis

UV-Vis characterization is one of the most frequent characterizations used to determine the AgNPs formation. The presence of chromophores in the organic components that make up AgNPs affect visible light in a way that can be measured. The following figure (Figure 2C) shows the UV-Vis characterization of AgEADN and AgEMDN from green synthesis. The wavelength used is in the range of 300–800 nm. The results of the spectrophotometer analysis showed that the maximum absorption of AgEADN and AgEMDN were 405 and 410 nm, respectively (Figure 2C). These results are in line with the previous study, which states that AgNPs absorb at a wavelength of 410 nm [10,18].

#### 2.3.2. FT-IR Analysis of AgAENPs and AgMENPs

AgAENPs and AgMENPs were characterized using FTIR spectroscopy to determine which functional group from the biomolecules are involved in reducing ion silver (Ag^+^) to silver nanoparticle (Ag^0^). In this study, the functional groups from the samples of patchouli leaf extract used as a bioreactor for synthesizing silver nanoparticles were also assessed. The most dominant group detected in all samples was the hydroxyl group (-OH). This shows that the synthesis of AgNP involves -OH groups as one of the groups that reduces Ag^+^ ions into Ag^0^. The FTIR results showed that almost all of the compounds in the crude extract were directly involved in the reduction process of silver ions in AgNO_3_. Amino acids were also reported as components that act as capping agents to stabilize silver nanoparticles [9,10]. The difference in AgNO_3_ concentration does not affect the number of groups involved but only decreases the level of silver ion, which is reduced biologically. The FT-IR spectrums of AgAENPs and AgMENPs can be seen in Figure 2B.

#### 2.3.3. XRD Analysis

The XRD analysis was performed to determine the crystallinity of AgAENPs and AgMENPs. The results of the AgAENPs and AgMENPs XRD analysis are presented in Figure 2D. XRD analysis showed that the tesla values (2θ) of 43.5°; 49.5° and 68.3° correspond to (111), (200), and (220). The characteristics of the AgNPs that were obtained are (111), (200), and (220) which also correspond to the Joint Committee on Powder Diffraction Standards (JCPDS) database. This indicates that the resulting AgNPs had a crystalline pattern. These results follow those reported by previous researchers that the characteristics of the XRD diffraction patterns for AgNPs are (111), (200), (220) [9,10].

#### 2.3.4. SEM Analysis

AgAENPs and AgMENPs were characterized by SEM morphology. It is well known that the formed AgNPs are agglomerated. The AgNPs that are produced typically have a particle form cluster morphology. The following figure shows the morphology of AgAENPs and AgMENPs (Figure 5).

The agglomeration of AgNPs using green synthesis method had been previously reported by those using the leaves extracts of *Mentha piperita* [7], *Calotropis gigantea* [9,19]. The characterization from SEM showed that AgMENPs gave the smallest average particle size with a diameter of 83.83 nm. The average diameter of AgEADN dan AgMENPs can be seen in Table 5.

### 2.4. Polyvinyl Alcohol/Corn Starch/Patchouli Oil Hydrogel Films Loaded with Silver Nanoparticles (PVA/CS/PO/AgNPs)

#### 2.4.1. Organoleptic Assay

The organoleptic assay of the hydrogel films obtained are presented in Table 6. In this study, the organoleptic assay included scent, color, shape, and homogeneity.

The results showed that F_0_ formula was white in color, this could be derived from corn-starch (CS). F_1_ and F_3_ produced the purple-colored film; the addition of silver nanoparticles (AgAENPs and AgMENPs) during the hydrogel preparation process is thought to be the source of this color. The formula of F_2_ and F_4_ gave brown color; this color was thought to be caused by patchouli oil (LFoPO and HFoPO) added to the hydrogel film preparations (Figure 6).

Overall, all the films obtained were homogeneous with semi-solid in shape. The results also show that the resulting film was flexible and easy to fold (flexible/foldable), does not crack (crack-free), and was free of air bubbles (bubble-free). This result was in line with the film produced by Mojally et al. [7].

#### 2.4.2. pH and Spreadability

The pH value of the hydrogel film corresponds to the pH of the skin, that being around pH 4.5–6.5 [20,21]. A lower pH can irritate the skin, whereas a higher pH can cause dryness of the skin and can induce skin irritation such as itching, rash, redness, and noise if continued [22].

The spreadability of the hydrogel films as measured longitudinally and transversely and was carried out for each addition of 1 g to 9 g of load. The results of the power spread test show that the higher the added load, the higher the power spread. The results also showed that the spreadability of films ranged from 1.5–2.90 cm. In this assay, only F_4_ (with HFoPO) shows a good spreadability i.e., 2.90 ± 0.14 cm. The pH and spreadability of the hydrogel films are presented in Table 7 behind.

#### 2.4.3. Gel Fraction and Swelling Index Test

Gel fraction is the degree of cross-linking formed in the structured hydrogel polymer network. The hydrogel is dried in a state of optimum swelling to determine the remaining fraction. The number of fractions not dissolved indicates the number of crosslinks formed. The gel fraction test was carried out to show the number of crosslinks formed. The value of the gel fraction was inversely proportional to the hydrogel swelling ability.

If too many cross-ties form this will decrease the gel’s swelling ability due to the reduced volume or space for storing absorbed water. The sample with the highest and lowest gel fraction values was F_0_ and F_2_, with the percentage yield of the gel fraction 2.96% and 1.88%, respectively (Table 8). In this study, all hydrogels films were weighed and soaked in aquadest for 24 h. A swelling ability study was conducted to understand the potential of liquids absorption ability in the wound site [23]. In this study, the hydrogels were weighed and soaked in PBS pH 7.4 for two days. Table 8 and Figure 7D show that all the formula’s swelling ratios were 1.07–1.50. The swelling ability is related to the hydrophilic character that increases the porosity and surface area of the network which leads to improving the hydrogel film’s swelling ratio [24].

#### 2.4.4. FT-IR Analysis

The Fourier transform infrared (FTIR) analysis was used to identify functional groups suspected to be involved in the hydrogels film. The FT-IR spectrum of all of hydrogel formula are presented in Figure 8.

The FTIR spectrum of hydrogel formula showed absorption bands at 3000–3600 cm^−1^ (OH), 2800–2900 cm^−1^ (-CH_2_, -CH_3_ str), 1738 cm^−1^ (-C=O), 1649 cm^−1^ (-C=C-), 1300–1400 cm^−1^ (-CH- bending), and 1107 cm^−1^ (-C-H-). These absorption bands are typical for the functional groups present in PVA, CS and Patchouli oil. However, in all these spectra, the absorption bands were very broad due to inter and intramolecular hydrogen bonding typically between the hydroxyl functional groups of PVA, CS, PO, and *Pogostemon cablin* Benth [25,26].

#### 2.4.5. SEM Analysis

Figure 7A shows a scanning electron microscopy (SEM) analysis of hydrogel films F_0_, F_1_, F_2_, F_3_, and F_4_. The analysis of SEM micrographs of formulas F_0_ to F_3_ revealed that all of the films produced had no cracks or pinholes and experienced little agglomeration, except for hydrogel film F_4_, which was more homogeneous and did not experience agglomeration (not agglomerated) between the matrix and silver nanoparticles. Figure 7B demonstrates that the hydrogel films F_3_ and F_4_ produced a slightly oily film compared to the hydrogel films F_1_ and F_2_ due to the addition of a high fraction and light fraction of patchouli oil (HFoPO and LFoPO, respectively).

The results also show that the phosphate buffer-soaked film produces a dry film (Figure 7D) compared to the water-soaked film (Figure 7C). Overall, the resulting hydrogel film was easily peeled off from the petri dish after drying at room temperature for 24 h.

#### 2.4.6. TGA and DTA Analysis

TGA and DTA spectra were recorded using a simultaneous thermal system in the temperature range from room temperature to 700 °C (Shimadzu, Kyoto, Japon, DTG-60). A ceramic (Al_2_O_3_) crucible was used for heating, and measurements were carried out in a nitrogen atmosphere (N_2_) at a heating rate of 10 °C/min. TGA and DTA curves of the hydrogel films are given in Figure 9.

TGA analysis of all hydrogel films reveals the same degradation pattern up to 60 °C, where entrapped moisture occurs. Aside from that, the results show that films almost lose weight at temperatures below 300 °C but gain weight above 300 °C. The degradation pattern of all formulas is nearly identical. The results showed that the hydrogel films F_2_ and F_4_ which were added with AgMENPs had the highest thermal stability compared to those added with AgAENPs, F_1_, and F_3_. This can be explained by the fact that silver nanoparticles (AgNPs) synthesized using methanol extract (MPLE) have smaller particle sizes (Table 4 and Figure 10). In comparison, the hydrogel film F_0_ (control) has a thermal stability between F_1_, F_3_, and F_2_; F_4_. DTA plot displays an intense exothermic peak between 200 °C and 300 °C. DTA profiles show that the sample’s complete thermal decomposition and crystallization coincide.

#### 2.4.7. Antimicrobial Activity

The disk diffusion method using different samples (filter paper and hydrogel) showed different inhibition zones, as in Table 9.

The zones of inhibition were measured in millimeters (mm). The positive control (Vancomycin) for *S. aureus* showed a 21.06 ± 0.33 mm zone of inhibition. F_1_ (with AgAENPs and LFoPO) showed a high zone of inhibition for *S. aureus* (12.13 ± 0.92 mm) while F_2_ (with AgMENPs and LFoPO) showed a medium effect (11.0 ± 0.4 mm), and patchouli oil exhibited the lowest inhibition. Similarly, F_3_ (with AgAENPs and HFoPO) showed a medium zone of inhibition effects (11.20 ± 0.52 mm), while F_4_ (with AgMENPs and HFoPO) showed a medium effect (11.0 ± 0.34 mm).

Compared to *S. epidermidis*, the positive control (Vancomycin) showed an 18.92 ± 1.23 mm zone of inhibition. F_1_ (with AgAENPs and LFoPO) showed a zone of inhibition for *S. epidermidis* (10.14 ± 2.92 mm), while F_2_, F_3_, and F_4_ showed low effects (8.0 ± 0.02 mm; 7.80 ± 0.14 mm; 7.90 ± 0.02 mm respectively). In this assay, both F_0_ and negative control showed no activity against *S. aureus* and *S. epidermidis*.

Overall, the hydrogel film of polyvinyl alcohol/corn starch/patchouli oil/hydrogel films loaded with silver nanoparticles showed better activity against *S. aureus* compared with *S. epidermidis* (Figure 10).

Regarding to this result, we have analyzed the SEM-EDX of hydrogel film F_1_ to determine the percentage of silver atom in the F_1_, and the result revealed that F_1_ contain silver atom with the percentage of atom and mass were 0.1% and 0.4% respectively (Figure 11).

## 3. Discussion

### 3.1. Phytochemistry Screening of Pogostemon cablin Benth

Table 1 exhibited that aqueous patchouli leaves extract (APLE) contained alkaloids, tannins, and terpenoids, while methanolic patchouli leaves extract (MPLE) contained alkaloids, flavonoids, tannins, and terpenoids. The previous study mentioned that extracts from leaves and stems of *P. cablin* Benth mainly contained patchouli alcohol, α-patchoulene, β-patchoulene, α-bulnesene, seychellene, norpatchoulenol, pogostone, eugenol and pogostol [3,14]. *P. cablin* Benth also contains volatile and non-volatile compounds. The volatile compounds generally consisted of patchouli alcohol, seychellene, α-guaiene, δ-guaiene, δ-patchoulene, ß-patchoulene, and pogostone, with the percentage of patchouli alcohol being 31.86% [2,14,15]. Our finding showed that aqueous patchouli leaf extract (APLE) and methanolic patchouli leaf extract (MPLE) contained of patchouli alcohol contents of 71.36% and 30.91% respectively (Figure 3A,B).

### 3.2. Patchouli Oil

In this study, the patchouli oil obtained from the steam distillation was fractionated using molecular distillation. The aim of the fractionation is to increase the content of the patchouli alcohol in the patchouli oil. In the global market, the quality of the patchouli oil is strongly influenced by the content of patchouli alcohol, where the higher the content of patchouli alcohol, the higher the quality and price of patchouli oil. By using a molecular distillation technique, our research has succeeded in obtaining light fraction of patchouli oil (LFoPO) and high fraction of patchouli oil (HFoPO) (Figure 1B and Table 1). The GC-MS analysis showed that, the percentage of patchouli alcohol contained in LFoPO and HFoPO were 28.68% and 60.66% respectively (Figure 4). Table 3 also showed that the percentage loss of the oil during the fractionation process of the patchouli oil was 5%. This finding indicates that economically the percentage of oil loss during the fractionation process is not indemnifying.

### 3.3. Green Synthesis Silver Nanoparticle

Green synthesis is a method of using plant extracts or biological material such as microorganisms for biosynthesis of nanoparticle. This approach is widely used by researchers due to it being considered cheap, easy, and environmentally friendly. Previous researchers reported that the biomaterial used in this method played the role of a bioreactor for the synthesis of nanoparticles. As bioreactor, plant or other biomaterials are able to transform metal ions [M^+^] into metal nanoparticles [M^0^] through the reductive capacities of the metabolites or the nucleation process such as alkaloids, steroids, tannins, polyphenol, terpenoids, and flavonoids) present in the plants. For example, *Brassica juncea*, *Medicago sativa*, and *Pleurotuscornucopiae var. citrinopileatus* were reported to be able to accumulate silver nanoparticles in the range of 50–100 nm [27].

In this study, the silver nanoparticles (AgNPs) was biosynthesized using aqueous (MPLE) of *Pogostemon cablin* Benth. The silver nanoparticles produced from APLE and MPLE were labeled as AgAENPs and AgMENPs respectively. Color changing occurred and acted as an indicator of the Ag^+^ → Ag^0^ reaction (Figure 1C,D). The solution of AgAENPs produced a blackish-brown solution, while AgMENPs produced a fine-light yellow brown solution. The previous reports had shown the color changing into either yellow-brown or dark brown as an indicator of AgNPs [9]. The color changing between AgAENPs and AgMENPs may be due to the role of secondary metabolites (such as alkaloids, flavonoids, tannins, and terpenoids) contained in APLE and MPLE (Table 1).

Our study found that silver nanoparticles synthesized in aqueous of patchouli leave extract (AgAENPs) and in methanolic of patchouli leave extract (AgMENPs) were able to accumulate silver nanoparticles in sizes of 167.70 ± 1.92 nm and 83.83 ± 2.94 nm respectively (Table 5). This finding showed that methanolic extract was able to accumulate smaller sized silver nanoparticle than the aqueous extract. Our results also proved that the methanolic extract was able to dissolve more secondary metabolites than the aqueous extract (Table 1).

#### Characterization of Silver Nanoparticles (AgAENPs and AgMENPs)

The UV-Vis spectrophotometer is one of the most common characterizations used to determine the AgNPs formation during synthesis or green synthesis. Our study revealead that the optimum absorptions of AgAENPs and AgMENPs were 405 and 410 nm, respectively (Figure 2C). Our results do not deviate much in comparison with those previously reported [5,9,10,18]. As a comparison, the previous study showed that the green synthesis of silver nanoparticles using chitosan and *C. gigantea* flower as reducing agent had absorption peaks at 410 nm and 430 nm, respectively [9,10]. Another study reported that AgNPs produced from *C. gigantea* from Saudi Arabia, revealed optimum absorption at 450 nm [9].

The FT-IR analysis was performed to determine which functional group from APLE and MPLE was involved in the formation of silver nanoparticles (Ag^0^). Figure 2B showed that hydroxyl group (-OH) was detected in both AgAENPs and AgMENPs. This result exhibited that hydroxyl group (-OH) from polar compounds (such as flavonoids and alkaloids) was one of the functional groups involved in formation of AgNPs. It was also believed that the -OH functional group plays an important role as a capping and reducing agent to stabilize AgNPs [3,9,19]. Figure 2B also showed that both AgAENPs and AgMENPs have similar spectrum patterns, where AgAENPs have a low intensity. This might be due to AgAENPs containing fewer metabolites than AgMENPs (Table 1). Figure 2B showed the spectrums profiles of both AgAENPs and AgMENPs. The results showed that both AgAENPs and AgMENPs have broad intermolecular bonded O-H_stretching_, medium C=C_stretching_, strong C=O_stretching_, and medium C-O_stretching_ vibrations at 3550–3200 cm^−1^; 2150 cm^−1^; 1685–1660 cm^−1^; and 1150–1085 cm^−1^, respectively.

The XRD analysis showed that the Number of Bragg reflections with tesla value (2θ) of 38.21°; 43.80° and 57.48° correspond to (111), (200), and (220) (Figure 2D). This indicated that the silver nanoparticles obtained were consistent with the crystalline pattern of silver. This result is correlated with previous studies [10].

The morphology of silver nanoparticles by SEM analysis showed that both AgAENPs and AgMENPs have particle form clusters. Interestingly, the SEM analysis showed that AgMENPs are uniform across their surface, being lighter, inclined to be obvious, and able to prevent the tendency to agglomerate compared with AgAENPs (Figure 5). This finding showed that the methanolic patchouli leave extract (MPLE) produces more secondary metabolites that then act as stabilizing and reducing agents in the synthesis of silver nanoparticle (as shown in Table 2). Besides that, SEM analysis also showed that AgMENPs synthesized using methanol as solvent gave the smallest average particle size with a diameter of 83.83 nm. Meanwhile, the silver nanoparticles synthesized using aqueous patchouli leave extract (APLE) gave a higher average size of silver nanoparticles with a diameter of 167.70 nm (Table 5). Table 5 shows that the silver nanoparticles synthesized using MPLE were effective in controlling the size of silver nanoparticles.

### 3.4. Polyvinyl Alcohol/Corn Starch/Patchouli Oil Hydrogel Films Loaded with Silver Nanoparticles (PVA/CS/PO/AgNPs)

In this study, the hydrogel films were made from polyvinyl alcohol (PVA), cornstrach (CS), patchouli oil (PO), glutaraldehyde (GL) and Tween-80 (TW-80). All the materials used in preparation of the film were safe and environmentally-friendly. The solvents used in the synthesis of the films were water and methanol. The nano-composite on the films was obtained by a green synthesis method using *Pogostemon cablin* leave extract namely, APLE and MPLE. In this assay, the hydrogel films were prepared in five formula (F_0_; F_1_; F_2_; F_3_; and F_4_) with different concentrations of patchouli alcohol (LFoPO and HFoPO) and nanocomposite of silver naoparticles (AgAENPs and AgMENPs).

In this study, the organoleptic assay of the films such as scent, color, shape, and homogeneity were also determined, and the results showed that all the films obtained were homogeneous and semi-solid in shape. The results also showed that formulas F_1_ and F_3_ produced purple-colored film due to colloidal of the silver nanoparticles solution (AgAENPs and AgMENPs), while formulas F_2_ and F_4_ produced brown color films, which could be caused by the addition of patchouli oil (LFoPO and HFoPO) during synthesis (Figure 6 and Table 5).

The pH and spreadability of the films were also determined and the results showed that the pH of the films corresponded to the pH of the skin (pH 4.5–6.5). The spreadability of the films showed that all the films were in the range of 1.5–2.90 cm (Table 6). The spreadability assay of the films was important to determine the ability of the films to spread on the skin’s surface. When a film is difficult to spread or too spreadable it will reduce the comfort level of use and the effectiveness of the film. If a film is too watery then that can cause the adhesive power of the film to decrease, and the contact time of the active substance with the skin is also reduced.

The gel fraction of the prepared hydrogel film was obtained in the range of 1.88–2.96% (Table 7). The gel fraction shows the extent to which the polymer chains have cross-linked with each other. It was found that the films with the oils loaded with silver nanoparticles gave decreasing gel fraction valuse. The swelling index of the hydrogel films was found in the range of 1.07–1.50 (Table 7). The swelling ratio of the hydrogel films is an essential parameter for mucoadhesive phenomena. Swollen polymers increase surface contact and mechanical entanglement for hydrogen bonding interactions between the polymer and mucous networks [28].

FT-IR analysis showed that all formulas have similar pattern on the FT-IR spectrums. All the formulas showed absorption of -OH functional group at wavelength of 3000–3600 cm^−1^; C-C bond at 2800–2900 cm^−1^; C=O bond at 1738 cm^−1^; C=C bonds at 1649 cm^−1^; -CH bonds at 1300–1400 cm^−1^ and -CH-bond at 1107.14 cm^−1^.

SEM analysis showed that all the films were homogenous and uniform, and the films also had no pinholes and cracks (Figure 7A). The drawback of this hydrogel film formula is that the resulting film is less transparent than the base (F_0_); this is due to the inclusion of patchouli oil (light and high fractions) and silver nanoparticles (AgAENPs and AgMENPs) in the hydrogel films F_1_ to F_4_, which are dispersed throughout the film.

The DTA analysis, like the TGA analysis, revealed that the hydrogel films F_1_, F_3_, F_2_, and F_4_ had an intense exothermic peak at temperatures ranging from 200 °C to 300 °C primarily attributed to the crystallization of silver nanoparticles (AgNPs). However, the F_0_ film (control) experienced an exothermic peak near 400 °C and continued to rise above 500 °C. DTA profiles show that the sample undergoes complete thermal decomposition and crystallization simultaneously (Figure 9).

The antibacterial assay was performed using disc diffusion method with a paper disc and hydrogel film. The diameter inhibition zone of each sample against *S. aureus* and *S. epidermidis* is listed in Table 8. In this assay, we used Vancomycin (30 µg) and Amoxicillin (25 µg) as positive controls for *S. aureus* and *S. epidermidis*, respectively. The positive control Vancomycin for *S. aureus* and Amoxicillin for *S. epidermidis* showed 21.06 mm and 18.92 mm diameter inhibition zones, respectively. The results in Table 8 show that the hydrogel film of F_1_ had a higher activity against both *S. aureus* and *S. epidermidis*. The extended studies demonstrated that the polyvinyl alcohol/corn starch/patchouli oil/hydrogel films loaded with AgNPs showed potential for wound healing because of the dispersion of nano-sized silver nanoparticles. The addition of silver nanoparticle (AgAENPs and AgMENPs) and patchouli oil (LFoPO and HFoPO) in hydrogel film increased the effect, further inhibiting the growth of both *S. aureus* and *S. epidermidis* compared with control. The results also showed that the hydrogel film obtained have strong activity against *S. aureus*. Our studies demonstrated that the PVA/CS/PO loaded with silver nanoparticles exhibited potential for antibacterial activity.

## 4. Materials and Methods

### 4.1. Materials

The ingredients used were *Pogostemon cablin* Benth, the light fraction of patchouli oil (PO) and a high fraction of patchouli oil, AgNO_3_, corn starch (CS), polyvinyl alcohol (PVA), deionized water, tween-80, ethanol, water, methanol and glutaraldehyde (GTD). The samples of patchouli leaves (*Pogostemon cablin* Benth) were collected from Nino Park, Atsiri Research Center, and Universitas Syiah Kuala. The samples were gathered using a purposive sampling technique, or non-random sampling, to identify the unique characteristics of the leaves. The patchouli leaves that were chosen were pest-free, perfectly shaped, and healthy patchouli leaves.

### 4.2. Preparation of Patchouli Leaf Extract

#### 4.2.1. Aqueous Patchouli Leaves Extract (APLE)

To create patchouli leaf simplicia, clean patchouli leaves were first dried at room temperature and then homogenized in a mixer. The produced simplicia was mixed with distilled water (10% *w*/*v*) and brought to a boil for five minutes. The extract was then filtered using Whatman Number 1 filter paper (size 90 mm) after being cooled to room temperature. The aqueous patchouli leaf extract was the name given to the resulting filtrate (APLE). The resulting APLE was then chilled and stored in a dark bottle for subsequent analysis [7].

#### 4.2.2. Methanolic Patchouli Leave Extract (MPLE)

The patchouli leaf simplicia that resulted was macerated in methanol (25% *w*/*v*) for 24 h at room temperature. The filtrate was created by filtering the resulting macerate through Whatman No. 1 filter paper. The filtrate obtained was labeled as methanolic patchouli leaf extract (MPLE). The resulting MPLE was refrigerated in a dark-colored bottle for further analysis [7].

#### 4.2.3. Phytochemicals Aqueous Patchouli Leave Extract (APLE) and Methanolic Patchouli Leaves Extract (MPLE) Screening

APLE and MPLE phytochemical screening was carried out to provide an overview of the compounds contained in the patchouli leaf plants. The phytochemical screening included examining the alkaloids, steroids, triterpenoids, saponins, tannins and polyphenols, glycosides, and flavonoids that were present within the plants [14].

### 4.3. Preparation of Light Fraction Patchouli Oil and Heavy Fraction Patchouli Oil

Patchouli oil obtained from the Atsiri Research Center (ARC) of Universitas Syiah Kuala was added to MgSO_4_ to bind the water molecules, then fractionated using a rotary evaporator to produce a heavy fraction of patchouli oil (patchouli alcohol content > 40%) and the light fraction of patchouli oil (patchouli alcohol content < 30%). The fractionation process aims to purify and improve the quality of patchouli oil. In this study, the redistillation process was carried out in 2 stages to produce a light fraction of patchouli oil (LFoPO) and a high fraction of patchouli oil (HFoPO) [29].

#### 4.3.1. FT-IR Analysis

The resulting of APLE, MPLE, LFoPO, and HFoPO samples were analyzed for the content of the functional groups produced using Fourier-Transform Infrared Spectroscopy (FT-IR spectroscopy). The FT-IR spectrum analysis was performed at wave numbers 400–4000 cm^−1^ [30].

#### 4.3.2. GC-MS Analysis

APLE, MPLE, LFoPO, and HFoPO samples were taken in amounts as much as 1 mL and injected into a gas chromatography-mass spectroscopy, GC-MS (Shimadzu GC-MS-QP2010 type). Operational conditions of GC-MS spectroscopy: The GC–MS instrument equipped with an RXI-5MS fused silica column (ID 30 m × 0.25 mm, film thickness 0.25 m) was operated under the following conditions: helium carrier gas (99.999%) at a flow rate of 1.0 mL/min and a separation ratio of 1/10. The column temperature programming was increased from 50 °C (sustained for 3 min) to 100 °C at a rate of 50 °C/min, from 100 to 200 °C at 8 °C/min, and then to 290 °C (continued for 10 min) at 100 °C/min. Injector and interface temperatures were 280 and 230 °C respectively, with pressure at 117.6 kPa, total flow at 25.0 mL/min, column flow at 2.0 mL/min, the linear velocity at 51.3 cm/s and purge flow at 3.0 mL/min [31].

### 4.4. Green Synthesis Silver Nanoparticles Using Patchouli Leaf Extract

One milliliter of APLE and MPLE were added to 9 mL of 0.1 M AgNO_3_ solution. After that, the mixture was given two hours to react at room temperature. An alteration in the mixture’s color from green to dark brown indicated the reaction that produces silver nanoparticles (AgNPs). Solutions of silver nanoparticles (AgNPs) produced from APLE, and MPLE is expressed as silver aqueous extract nanoparticles (AgAENPs) and silver methanolic extract nanoparticles (AgMENPs), respectively [7].

#### 4.4.1. UV-Vi Spectrophotometer Analysis

UV-Vis spectrophotometer analysis of AgAENPs and AgMENPs was carried out at a wavelength range of 300–800 nm using a UV-Vis spectrophotometer (Shimadzu, UV 2500) [10].

#### 4.4.2. FT-IR Analysis

The silver nanoparticles formed (AgAENPs and AgMENPs) were analyzed using FT-IR spectroscopy (Cary 630 FTIR spectrometer, Agilent Technologies, 5301 Stevens Creek Blud, Santa Clara, CA, USA) with a wavelength of 500–4500 cm^−1^ [32].

#### 4.4.3. XRD Analysis

The formed silver nanoparticles (AgAENPs and AgMENPs) were placed on a glass substrate. XRD pattern (XRD-6000 Shimadzu, 7102 Riverwood Drive, Columbia, MD, USA) analysis was taken at room temperature with a Tesla angle (2θ) from 10° ≤ 2θ ≤ 70° [33].

#### 4.4.4. SEM-EDX Analysis

Micro morphology and size of silver nanoparticles (AgAENPs and AgMENPs) were analyzed using a Scanning Electron/X-ray Energy Dispersion Microscope (SEM-EDX, Sem-eds Carl ZeissBruker EVO MA 10, Carl Zeiss Microscopy, One North Broadway, White Plains, NY, USA) at 15 kV energy with 5000× and 10,000× magnification [9,10].

### 4.5. Polyvinyl Alcohol/Corn Starch/Patchouli Oil Hydrogel Films loaded with Silver Nanoparticles (PVA/CS/PO/AgNPs)

The formulation of PVA/CS/PO/AgNPs preparations was carried out based on the method developed by [7] with slight modifications. PVA/CS/PO/AgNPs formulations can be seen in Table 10. To prepare the F_1_ formula, 40 mL of distilled water was added to a container with 2.8 and 1.2 g of CS and PVA, respectively. After that, 12 mL of glycerin and 0.4 g of LFoPO combined with 4 mL of Tween 80 (TW-80) were added. After being continuously stirred for 10 min at 85 °C to achieve homogeneity, the mixture was set aside to cool to room temperature. The cooled preparations were then mixed with 1 mL of silver nanoparticle suspension resulting from green synthesis using aqueous patchouli leaf extract (AgEADN) while stirring using a magnetic stirrer for 1 h at room temperature to produce a homogeneous preparation. One milliliter of acidified glutaraldehyde solution was then added to the homogeneous preparation drop wise while stirring using a magnetic stirrer at a constant speed. Acidified glutaraldehyde (GL) aims to maintain the stability of hydrogel preparations. Another purpose of adding glutaraldehyde is as a stabilizing agent for silver nanoparticles because of its ability to crosslink with silver nanoparticles. The resulting hydrogen preparation was immediately poured into a petri dish and allowed to dry at room temperature to produce hydrogel films designated as F_1_. The same procedure is also used to prepare hydrogel film F_0_; F_2_; F_3_; and F_4_ preparations [7].

### 4.6. Evaluation of Hydrogel Film Preparations

#### 4.6.1. Fraction Gel

The resulting hydrogel films were weighed after 6 h of drying at 50 °C in the oven (*W_o_*). The gel was soaked in distilled water for 24 h, then dried again using an oven at 50 °C for 6 h and weighed. (*W_e_*) [20]. The gel fraction was then calculated using the formula:$$Gel\;fraction=\left(\frac{W_{e}}{W_{o}}\right)\times 100$$

#### 4.6.2. Swelling Index

The resulting hydrogel film was then cut into 1 cm × 1 cm squares and dried for 12 h at 60 °C (*W_a_*). The gel was then immersed in a simulated wound fluid solution, SWF (phosphate buffer solution, pH 7.4) at 37 °C for 24 h, and dried again using an oven at 60 °C for 12 h and weighed (*W_s_*) [34]. The gel fraction was then calculated using the formula:$$Swelling\;index=\left(\frac{W_{s}}{W_{a}}\right)\times 100$$

#### 4.6.3. pH Test

The pH of all hydrogel films (F_0_, F_1_, F_2_, F3, and F_4_) was measured using a digital pH meter. The pH test was repeated three times (triplicate) [35↔21].

#### 4.6.4. Spreadability Test

A total of 0.5 g of hydrogel film (1 cm × 1 cm) was cut from each formula and placed in a transparent glass cup. Then a transparent glass cup (cover glass) was placed on it, and weights (1, 3, 5, 7, and 9 g) were place on it for 60 s each. In addition, the spreading power of each formula was determined. The spreadability test was repeated three times (triplicate) [35].

### 4.7. Characterization of Hydrogels

#### 4.7.1. FT-IR Analysis

The interaction between several components in the preparation of all hydrogel film formulas was evaluated using FT-IR spectroscopy at wave numbers 4000–400 cm^−1^ [9,36].

#### 4.7.2. SEM Analysis

Morphological analysis of all hydrogel film formulations was performed using a scanning electron microscope (JEOL, Akishima, Tokyo, Japan) with an accelerating voltage ranging from 0.5 kV to 30 kV. The resulting gel morphology was then observed and analyzed [10,36].

#### 4.7.3. TGA and DTA Analysis

Determination of the degree of hydration by thermogravimetric analysis (TGA) and differential thermal analysis (DTA) of hydrogel film samples were carried out on a DTG-60 (Schimadzu, Kyoto, Japan) at a maximum temperature of 1100 °C. TGA and DTA analyses were performed in a nitrogen (N2) atmosphere at a heating rate of 10 °C/min to evaluate the thermal stability of the hydrogel films [7].

#### 4.7.4. Antibacterial Assay

The Kirby-Bauer disc diffusion method was used to test antibacterial activity. As much as 19 g of Mueller Hinton Agar (MHA) medium was added to 500 mL of sterile distilled water, which was then homogenized, and heated to boiling. After that, the media was sterilized in an autoclave for 15 min at 121 °C. The media was then poured into a petri dish in quantities of up to 25 mL and allowed to solidify. Then, using a cotton swab (0.1 mL), a suspension of *Staphylococcus aureus* or *Staphylococcus epidermidis* bacteria was applied to the entire surface of the MHA media. The disk that had previously been soaked in a positive control solution was then placed in the petri dish (Vancomycin 30 µg for testing against *Staphylococcus aureus* or Amoxicillin and 25 µg for testing against *Staphylococcus epidermidis*), aquadest as negative control and basis of hydrogel film (F_0_) as positive control. The diameter of the inhibition formed after incubating the media at 37 °C for 24 h was then measured. The clear zone formed on the media was measured to determine antibacterial activity.

## 5. Conclusions

The PVA/CS/PO/AgNPs hydrogel films obtained were flexible, easy to fold, free of holes and air bubbles and had no side reactions. The FTIR analysis showed the presence of hydrogen bonds between the functional groups of PVA, CS, and PO. The SEM analysis revealed that the hydrogel film was slightly agglomerated and did not exhibit cracking or pinholes. The analysis of pH, spreadability, gel fraction, and swelling index showed that the resulting PVA/CS/PO/AgNPs hydrogel films met the expected standards except for the organoleptic properties of the resulting colors, which tended to be slightly darker in color. The TGA and DTA analysis showed that the hydrogel films had the highest thermal stability and can be safely used up to 200 °C. The results showed that F_1_ containing percentage weight of silver (Ag) of 0.4% exhibited strongest antibacterial activity against both *S. aureus* and *S. epidermis* as determined by the disc diffusion method.

## Figures and Tables

**Figure 1 molecules-28-02020-f001:**
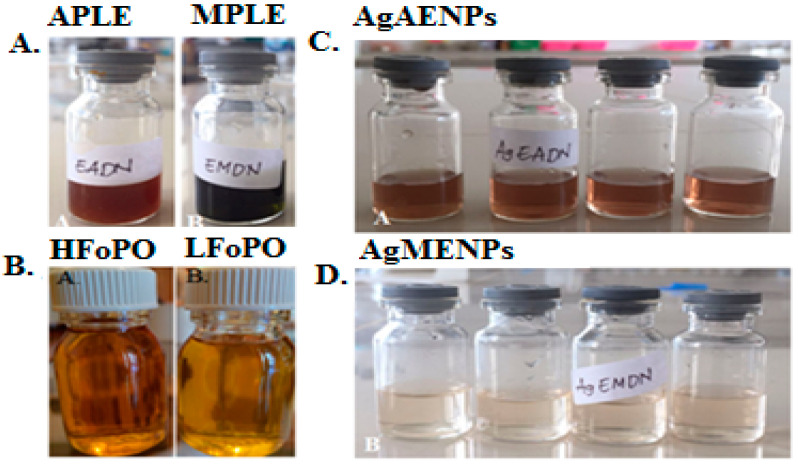
(**A**) Aqueous patchouli leaves extract (APLE) and methanolic patchouli leaves extract (MPLE); (**B**) High fraction of patchouli oil (HFoPO) and light fraction of patchouli oil (LFoPO); (**C**) silver nanoparticles synthesized in methanolic of patchouli leave extract (AgAENPs); and (**D**) silver nanoparticles synthesized in aqueous of patchouli leave extract (AgMENPs).

**Figure 2 molecules-28-02020-f002:**
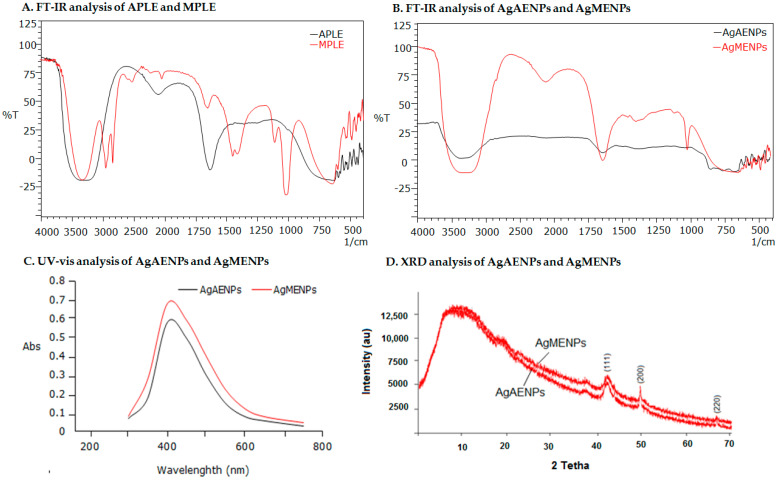
(**A**) FT-IR analysis of APLE and MPLE; (**B**) FT-IR analysis of AgAENPs and AgMENPs; (**C**) Spectrum UV-Vis analysis of AgAENPs and AgMENPs; (**D**) XRD analysis of AgAENPs and AgMENPs.

**Figure 3 molecules-28-02020-f003:**
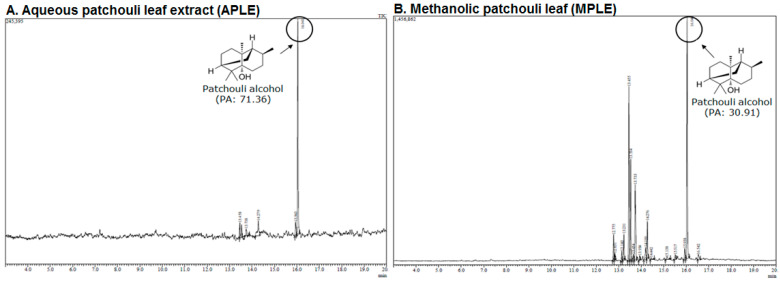
The GC-MS spectrum; (**A**) aqueous patchouli leaves extract (APLE), and (**B**) methanolic patchouli leaves extract (MPLE).

**Figure 4 molecules-28-02020-f004:**
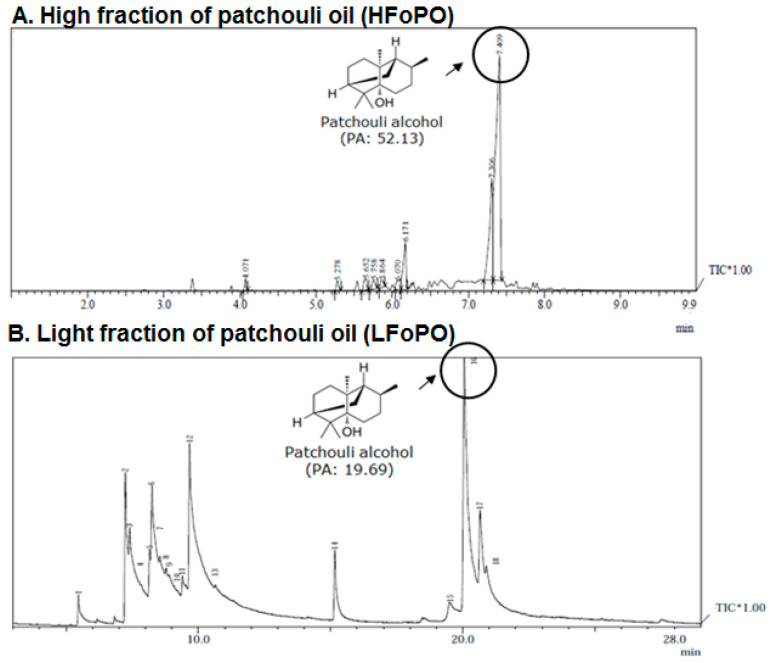
The GC-MS spectrum of patchouli oil; (**A**) High fraction of patchouli oil (HFoPO) and (**B**) Light fraction of patchouli oil (LFoPO).

**Figure 5 molecules-28-02020-f005:**
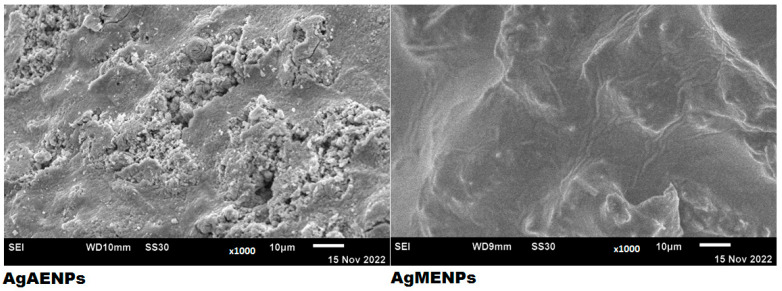
SEM analysis of AgAENPs and AgMENPs resulting from the green synthesis.

**Figure 6 molecules-28-02020-f006:**
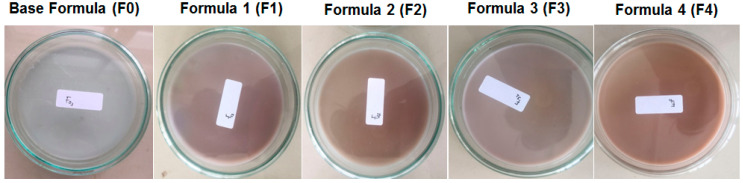
Organoleptic properties of polyvinyl alcohol/cornstarch/patchouli oil/hydrogel films loaded with silver nanoparticles.

**Figure 7 molecules-28-02020-f007:**
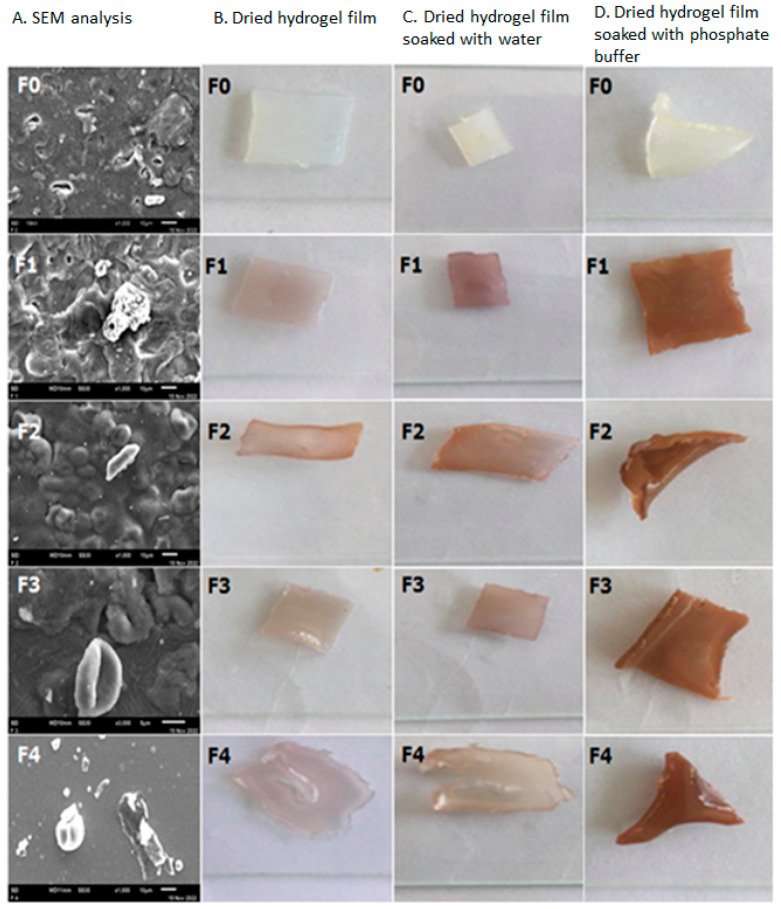
SEM analysis of Polyvinyl alcohol/Corn starch/Patchouli oil Hydrogel films (**A**) loaded with silver nanoparticles (F_0_; F_1_; F_2_; F_3_; and F_4_); (**B**) Dried hydrogel; (**C**) Dried hydrogel soaked with water, and (**D**) Dried hydrogel soaked with phosphate buffer.

**Figure 8 molecules-28-02020-f008:**
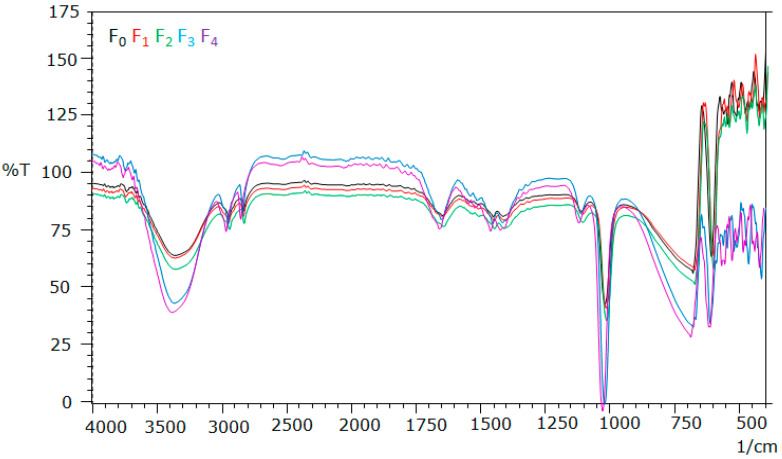
Analysis of FT-IR of Polyvinyl alcohol/Corn starch/Patchouli oil/Hydrogel films loaded with silver nanoparticles (F_0_; F_1_; F_2_; F_3_; and F_4_).

**Figure 9 molecules-28-02020-f009:**
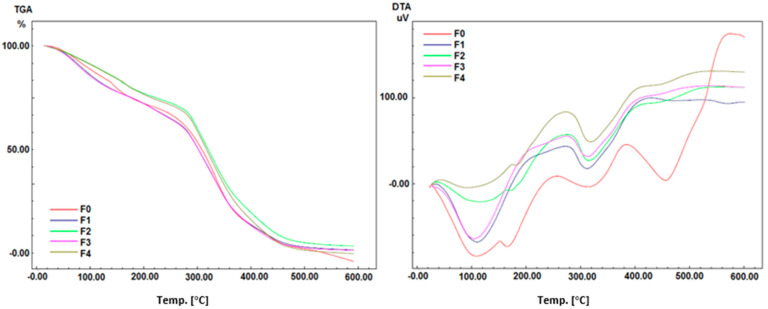
Analysis of TGA and DTA of Polyvinyl alcohol/Corn starch/Patchouli oil/Hydrogel films loaded with silver nanoparticles (F_0_; F_1_; F_2_; F_3_; and F_4_).

**Figure 10 molecules-28-02020-f010:**
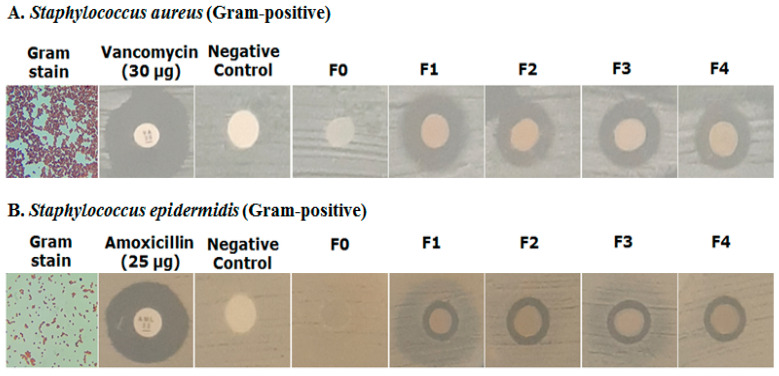
Antimicrobial activity test of hydrogel film preparation against *S. aureus* and *S. epidermidis*.

**Figure 11 molecules-28-02020-f011:**
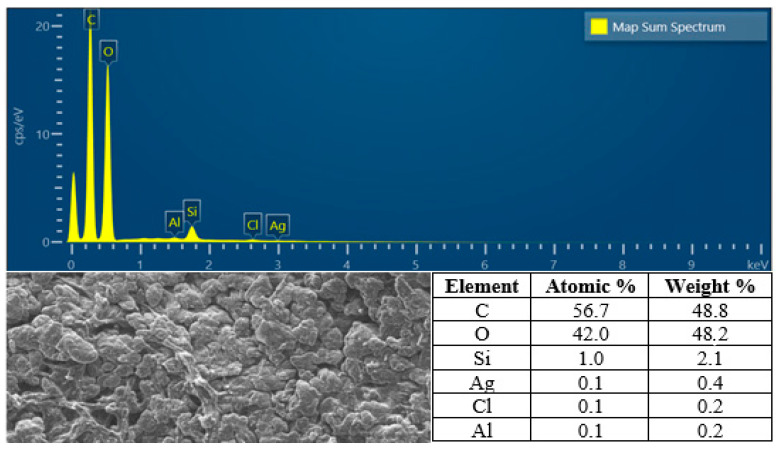
The SEM-EDX analysis of F_1_.

**Table 1 molecules-28-02020-t001:** Phytochemicals Aqueous patchouli leaf extract (APLE) and methanolic patchouli leaf extract (MPLE) Screening.

Extract	Secondary Metabolites				
	Alkaloids	Flavonoids	Tannins	Saponins	Terpenoids
APLE	(ve)+	(ve)−	(ve)+	(ve)−	(ve)+
MPLE	(ve)+	(ve)+	(ve)+	(ve)−	(ve)+

Note: (ve)+: positive for containing secondary metabolites; (ve)−: negative contains secondary metabolites.

**Table 2 molecules-28-02020-t002:** The characteristic peaks of FTIR analysis before and after the chemical reactions.

Before Chemical Reaction				After Chemical Reaction			
APLE		MPLE		AgAENPs		AgMENPs	
Wavelength (cm^−1^)	Functional Group	Wavelength (cm^−1^)	Functional Group	Wavelength (cm^−1^)	Functional Group	Wavelength (cm^−1^)	Functional Group
3400.22	-O-H	3350.15	-O-H	3590.12	-O-H	3450.14	-O-H
2138.15	-C-N	2850.14	-C-H			2200.11	-C-N
1640.20	-C=C	1680.22	-C=C	1675.11	-C=C	1670.21	-C=C
		1340.20	-C-N			1150.10	-C-O
		1050.45	-C-O				
		1125.16	-C-O				

**Table 3 molecules-28-02020-t003:** The chemical composition of aqueous patchouli leaf extract (APLE) and methanolic patchouli leaf extract (MPLE) by GC-MS analysis.

No.	Chemical Composition		Chemical Composition	
	APLE	% Area	MPLE	% Area
1.	α-Guaiene	8.25	4.7-Methanoazulene	2.97
2.	α-Patchoulene	2.36	Cyclohexene	0.86
3.	Azulen	10.67	Seychellene	1.16
4.	Veridiflorol	7.36	Caryophyllene	3.18
5.	Patchouli alcohol	71.36	α-Guaiene	20.44
6.			Seychellene	12.66
7.			α-Humulene	1.24
8.			α-Patchoulene	13.38
9.			Azulene	0.69
10.			α-Guaiene	2.40
11.			α-Bulnesene	4.87
12.			α-Panasinsen	0.53
13.			Caryophyllene-oxide	0.85
14.			Cyclopropenazulene	0.89
15.			Veridiflorol	1.71
16.			Patchouli alcohol	30.91
17.			Cyclohexanol	1.25
	Total	100.00	Total	100.00

**Table 4 molecules-28-02020-t004:** The percent yield of the light fraction and the heavy fraction of patchouli oil was produced using the molecular fractionation method.

Treatment	Operational Conditions				Patchouli Alcohol ^c^ (%)
	Feed Volume (mL)	Temperature (°C)	Result (mL)	Yield Percent (%)	
LfoPO ^a^	200	125	121	60.5	28.68
HfoPO ^b^	79	115–160	69	34.5	60.66
Loss	-	-	10	5	-

Note: ^a^ light fraction of patchouli oil (LFoPO); ^b^ high fraction of patchouli oil (HFoPO); ^c^ by GC-MS analysis.

**Table 5 molecules-28-02020-t005:** The diameter size of AgAENPs dan AgMENPs.

AgNPs	Average Diameter ± SD (nm)
AgAENPs ^a^	167.70 ± 1.92
AgMENPs ^b^	83.83 ± 2.94

^a^ AgAENPs = silver aqueous extract nanoparticles; ^b^ AgMENPs = silver methanolic extract nanoparticles.

**Table 6 molecules-28-02020-t006:** Organoleptic properties of polyvinyl alcohol/corn starch/patchouli oil hydrogel films loaded with silver nanoparticles.

Formula	Scent	Color	Shape	Homogeneity
F_0_	No scent	White	Semi-solid	Homogeneous
F_1_	No scent	Purple	Semi-solid	Homogeneous
F_2_	Patchouli scent	Brown	Semi-solid	Homogeneous
F_3_	No scent	Purple	Semi-solid	Homogeneous
F_4_	Patchouli scent	Brown	Semi-solid	Homogeneous

**Table 7 molecules-28-02020-t007:** Average pH test and spreadability of polyvinyl alcohol/corn starch/patchouli oil/hydrogel films loaded with silver nanoparticles.

Formula	pH		Spreadability (cm)									
	pH	SD	1(g)	SD	3(g)	SD	5(g)	SD	7(g)	SD	9(g)	SD
F_0_	6.03	±0.05	2.00	±0.00	2.15	±0.07	2.25	±0.07	2.25	±0.07	2.35	±0.07
F_1_	6.40	±0.10	1.50	±0.00	1.60	±0.00	1.75	±0.07	1.90	±0.00	2.00	±0.00
F_2_	6.40	±0.10	2.10	±0.14	2.30	±0.00	2.40	±0.00	2.55	±0.07	2.70	±0.00
F_3_	6.60	±0.15	1.70	±0.14	1.85	±0.07	1.95	±0.07	2.00	±0.00	2.25	±0.07
F_4_	6.50	±0.05	1.80	±0.14	2.35	±0.07	2.45	±0.07	2.50	±0.14	2.90	±0.14

**Table 8 molecules-28-02020-t008:** Gel fraction and swelling index test polyvinyl alcohol/corn starch/patchouli oil/hydrogel films loaded with silver nanoparticles.

Formula	Gel Fraction			Swelling Index		
	*W_o_* (g)	*W_e_* (g)	Yield (%)	*W_a_* (g)	*W_s_* (g)	Yield (%)
F_0_	0.032	0.095	2.96	0.123	0.132	1.07
F_1_	0.034	0.098	2.88	0.111	0.135	1.21
F_2_	0.016	0.030	1.88	0.110	0.120	1.09
F_3_	0.032	0.076	2.30	0.118	0.134	1.13
F_4_	0.040	0.088	2.20	0.093	0.140	1.50

**Table 9 molecules-28-02020-t009:** Antibacterial activity test of polyvinyl alcohol/corn starch/patchouli oil/hydrogel films loaded with silver nanoparticles.

Formula	Inhibition Zone		Formula	Inhibition Zone	
	*S. aureus*			*S. epidermidis*	
	(mm)	SD		(mm)	SD
Vancomycin (30 µg)	21.06	±0.33	Amoxicillin (25 µg)	18.92	±1.23
Aquadest	0.00	±0.00	Aquadest	0.00	±0.00
F_0_	0.00	±0.00	F_0_	0.00	±0.00
F_1_	12.13	±0.92	F_1_	10.14	±2.92
F_2_	11.00	±0.40	F_2_	8.00	±0.02
F_3_	11.20	±0.52	F_3_	7.80	±0.14
F_4_	11.00	±0.34	F_4_	7.90	±0.02

**Table 10 molecules-28-02020-t010:** Formulation of Polyvinyl alcohol/Corn starch/Patchouli oil Hydrogel Films loaded with Silver Nanoparticles (PVA/CS/PO/AgNPs) [7].

No.	Ingredients	Formula				
		F_0_	F_1_	F_2_	F_3_	F_4_
1.	*Corn starch*, CS (gram)	2.8	2.8	2.8	2.8	2.8
2.	*Polyvinyl alcohol*, PVA (gram)	1.2	1.2	1.2	1.2	1.2
3.	Aquadest (mL)	40	40	40	40	40
4.	Light fraction patchouli oil, LFoPO (gram)	-	0.4	0.4	-	-
5.	Heavy fraction patchouli oil, HFoPO (gram)	-	-	-	0.4	0.4
6.	Tween 80, TW-80 (0.4 g in 4 mL etanol)	4	4	4	4	4
7.	Glycerol (2 mL in 10 mL etanol)	12	12	12	12	12
8.	Silver nanoparticles, AgEADN (mL)	-	1	-	1	-
9.	Silver nanoparticles, AgEMDN (mL)	-	-	1	-	1
10.	*Acidified* glutaraldehida, GL ^a^ in 1 mL etanol (mL)	1	1	1	1	1

Note: ^a^: acidification is performed by adding HCl or KCl until a pH of 2.9 is obtained.

## Data Availability

Data that support the findings of this study are available from the corresponding author upon reasonable request.

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
