# Peer review of "Fabrication and Evaluation of Polyvinyl Alcohol/Corn Starch/Patchouli Oil Hydrogel Films Loaded with Silver Nanoparticles Biosynthesized in Pogostemon cablin Benth Leaves’ Extract"

_molecules, 2023, doi:10.3390/molecules28052020_

Round 1
Reviewer 1 Report
The article "Polyvinyl Alcohol/Corn Starch/Patchouli Oil Hydrogel Films Loaded with Silver Nanoparticles Biosynthesized in Pogostemon cablin Benth Leaves’ Extract: A Study for Patchouli Oil-silver Nano Diabetic Wound Dressing" focuses on preparation and antibacterial properties of PVA/CS/PO/AgNPs hydrogel films, which is a meaningful work. This article meets the requirements of this journal. In this paper, the composite hydrogel and its materials were characterized in detail by infrared, XRD, scanning electron microscopy, liquid chromatography, and so on. However, there are still some shortcomings in the text. Therefore, if the author can make minor changes to the manuscript according to the following suggestions, the article is publishable.
FS1: In the abstract section, the first occurrence of the word AgAENPs should be explained in detail.
FS2: The number of keywords is large, and I recommend narrowing it down to 3-5.
FS3: Most of the pictures in the article are not clear and beautiful enough. I suggest that they be modified.
FS4: The format of headers and illustrations should be uniform in the text.
FS5: I suggest polishing the language of the full text.
Author Response
The revision according to the reviewer's comments (Reviewer 1#) in the article are in green color

Reviewer 2 Report
1. Too many components in the designed hydrogel and the authors declared that these hydrogel films performed better in the potential application as patchouli oil-silver nano diabetic wound dressing, this conclusion is very confusing? according to what results? 2. Please describe the results in detail, but not provide all the data you have! 3. Too many abbreviations in this study, it is very hard to catch the meanings of the results derived from each component! 4. Please identify every characteristic peaks in FTIR analysis before and after the chemical reactions. 5. The authors have to delete the F1-F4 in the abstract, if you have defined them beforehand in abstract! Because the reader have to find them in Table 6. 6. Can not correlate the references to the contains in the manuscript!. 7. What is the difference between two SEM images in Figure 5, and what is its related contribution to the wounding healing? 8. The authors have to describe in detailed on each obtained results in discussion section! And indicate the related contribution of each component, for it is a complexed system in the present study. 9. Can the authors provide the TEM images of synthesized nanoparticles? How about its sphericity? 10. What is the thickness of the obtained membrane in Figure 8. It seemed too thick in clinical application! 11. The authors did not execute in vivo animal studies, how they declared that “The hydrogel films containing AgAENPs with patchouli 33 oil light fraction (LFoPO) performed better in the potential application as patchouli oil-silver nano diabetic wound dressing. 12. It is a labor work in this study, however, no evidences in the wounding healing I suggest the title should be corrected!.
Author Response
The revision according to the reviewer's comments (Reviewer 2#) in the article are in green color

Reviewer 3 Report
Polyvinyl Alcohol/Corn Starch/Patchouli Oil Hydrogel Films Loaded with Silver Nanoparticles Biosynthesized in Pogostemon cablin Benth Leaves’ Extract: A Study for Patchouli Oil-silver Nano Diabetic Wound Dressing by Khairan Khairan et al. is an original research article. The results presented in the form of tables and plots are interesting, however the authors resist to elaborate introduction, materials and methods, results and discussion and conclusions. The abstract is written properly.
The interesting part of the results presented is the comparison by GC/MS the two media of extraction of Patchouli leafs: water and methanolic. The problem seems to be the toxicity of methanol. Thus the water based extract seems to be a better choice despite less rich content o active compounds.
The concept of green synthesis is not clear to me. The authors should explain more in their paper this issue, cite some realted positions in the literature of the context.
The use of nanoparticles of silver as antibacterial ingredient of their compositions is a very good choice. All the characterisation methods, incluidng SEM/EDX are properly choosen and presented.
In summary, it is worth to write the whole article starting from introduction to conclusions.
Author Response
The revision according to the reviewer's comments (Reviewer 3#) in the article are in red color

Round 2
Reviewer 2 Report
This revised manuscript is corrected accordingly, I think it can be accepted in your journal!.
Reviewer 3 Report
The authors addressed all the comments, thus the article is recommended for publication in present form.